# Psychometric evaluation of the Persian nursing students' learning self-efficacy instrument

Elaheh Habibpour[1], Reza Nemati-Vakilabad[2], Alireza Mirzaei[3], Nargess Ramazanzadeh[2], Sevda Gardashkhani[4], Mehraban Shahmari[5]*

1 MSc of Medical-Surgical Nursing Student, Students Research Committee, School of Nursing and Midwifery, Ardabil University of Medical Sciences, Ardabil, Iran, 2 Department of Medical-Surgical Nursing, School of Nursing and Midwifery, Ardabil University of Medical Sciences, Ardabil, Iran, 3 MSc of Emergency Nursing, Department of Emergency Nursing, School of Nursing and Midwifery, Ardabil University of Medical Sciences, Ardabil, Iran, 4 Student Research Committee, Tabriz University of Medical Sciences, Tabriz, Iran, 5 Department of Medical-Surgical Nursing, School of Nursing and Midwifery, Ardabil University of Medical Sciences, Ardabil, Iran

* mehrabanshahmari@yahoo.com

## Abstract

The preparedness of the nursing workforce is a critical determinant of patient safety and healthcare quality, with academic performance serving as a key contributor to this preparedness. Self-efficacy plays a significant role in shaping the motivation and success of nursing students, necessitating the availability of valid and reliable instruments for its assessment. This study aimed to translate a multidimensional learning self-efficacy instrument for nursing students into Persian and evaluate its psychometric properties within the Iranian context. Employing a methodological design, the study recruited 450 nursing students through convenience sampling between September and December 2024. The instrument underwent translation/back-translation, with face validity (Impact Score > 1.5), content validity ratio (CVR = 0.8–1), and content validity index (CVI = 80–100%) established. Psychometric evaluation included construct, convergent, and discriminant validity, internal consistency, and test-retest reliability. Students from all academic years participated in construct validity testing through Exploratory Factor Analysis (EFA) and Confirmatory Factor Analysis (CFA) using SPSS 24.0 and AMOS 24.0. EFA identified a five-factor structure explaining 73.43% of variance, supported by strong sampling adequacy (Kaiser-Meyer-Olkin = 0.87) and significant Bartlett's test (p < 0.001). CFA confirmed model fit for the 21-item Nursing Learning Self-Efficacy (NLSE) instrument, with robust indices: Parsimonious Comparative Fit Index (PCFI = 0.75), Parsimonious Normed Fit Index (PNFI = 0.72), Incremental Fit Index (IFI = 0.93), and Comparative Fit Index (CFI = 0.93). Internal consistency (Cronbach's α = 0.93) and test-retest reliability (Intraclass Correlation Coefficient = 0.89) were excellent. These findings establish the Persian version of the NLSE as a valid and reliable self-report instrument. By providing a culturally adapted instrument, this study equips nursing educators and researchers

**Data availability statement:** All relevant data are within the manuscript and its Supporting Information files.

**Funding:** The author(s) received no specific funding for this work.

**Competing interests:** The authors have declared that no competing interests exist.

with a practical means to assess and enhance self-efficacy among nursing students, ultimately contributing to improved educational outcomes and a more competent nursing workforce capable of addressing Iran's evolving healthcare demands.

## Introduction

Sustainable challenges in patient safety and healthcare quality are closely tied to the preparedness of the nursing workforce [1,2]. The primary objective of nursing education is to cultivate professionals equipped with the essential knowledge and skills; any oversight in this training process can result in resource wastage and diminished service quality [3]. Research has demonstrated that the outcomes of nursing education have a significant impact on both the global workforce and national healthcare systems [4,5].

The factors that predict academic success or failure among nursing students are complex, including academic self-efficacy and clinical performance [6,7]. Academic decline, often stemming from inadequate cognitive preparedness, can lead to a shortage of nursing personnel, which carries substantial economic and social consequences. Such shortages burden existing staff and contribute to higher turnover rates [8–10].

Understanding the determinants of academic success, particularly self-efficacy, is essential [6]. According to Bandura's (1977) social cognitive theory, self-efficacy refers to an individual's perception of their ability to organize and execute specific tasks or actions required to achieve a particular type of performance [11]. Numerous studies have demonstrated that self-efficacy has a significant influence on students' motivation, cognition, actual performance, and confidence [12–14]. Furthermore, higher levels of self-efficacy are associated with better performance outcomes [15–17]. Individuals with higher self-efficacy are more likely to set ambitious goals, accept greater challenges, complete academic tasks more effectively, and utilize a broader range of learning strategies [18]. In the field of nursing education, self-efficacy also plays a critical role in students' academic success and serves as a significant predictor of their educational performance [19,20]. Recent studies have demonstrated that students' perceptions of their learning experiences can significantly enhance or undermine their learning self-efficacy [21,22]. Learning self-efficacy, which refers to students' judgments about their capabilities to achieve learning goals [18], is a key factor influencing learners' academic performance [23,24]. Given this context, there is a recognized need for a standardized instrument to assess nursing students' learning self-efficacy in Iran. Cheraghi et al. (2009) in Iran developed and tested the Self-Efficacy in Clinical Performance (SECP) scale, a psychometrically sound and valid instrument specifically designed to evaluate the self-efficacy of nursing students during clinical practice, including assessment, planning, implementation, and evaluation of care [25]. While the SECP instrument addresses clinical aspects of clinical nursing education, it excludes other learning domains, such as the practical application of theoretical knowledge, critical thinking, or communication skills. In addition, current

assessment tools predominantly focus on academic dimensions [18,26,27], underscoring the need for a multi-dimensional instrument that comprehensively evaluates self-efficacy. In contrast, the instrument used in this research—the Learning Self-Efficacy Scale for Nursing Students (NLSE), developed by Yin et al. (2024)—adheres to a multi-dimensional framework, covering academic and clinical learning self-efficacy, with the aim of a more comprehensive assessment of students' learning capacity in different areas. This instrument is a multidimensional tool to assess nursing students' learning self-efficacy, evaluating their confidence in applying nursing concepts, critical thinking skills, executing nursing tasks, and practical communication abilities [28].

The psychometric evaluation of this instrument in Persian could have a profound impact on enhancing educational processes and improving health quality within Iranian society. Additionally, the psychometric properties of this instrument, in conjunction with improving educational processes, can help predict and evaluate students' success in completing course-work and academic progress. This research represents a pivotal advancement for future studies on self-efficacy in nursing education in Iran and can significantly contribute to knowledge development in this area. Possessing a valid and reliable instrument will facilitate the early detection of educational needs, permit targeted intervention, and ultimately assist in improving both Iranian nursing education and patient care outcomes.

### Objectives

Due to the absence of a validated Persian instrument for measuring learning self-efficacy among nursing students, the present study proposes the translation and investigation of the psychometric properties of the multidimensional Learning Self-Efficacy Scale for Nursing Students developed by Yin et al. (2024) in the Iranian context. More specifically, the present study aims to assess the face, content, construct validity, and reliability (internal consistency and stability) of the Persian version of the instrument. The ultimate goal is to develop a culturally and linguistically valid instrument that will enable nursing faculty and researchers to measure and enhance students' learning self-efficacy, thereby contributing to improved educational outcomes and healthcare in Iran.

## Materials and methods

### Design and Setting

This study is a methodological study conducted using a cross-sectional approach. The objective of this study was to standardize and validate the Nursing Students' Learning Self-Efficacy (NLSE) instrument through a cross-sectional survey conducted at the School of Nursing and Midwifery in Ardabil, northwest Iran. The research seeks to assess whether the psychometric properties of the NLSE instrument, initially developed by Yin et al. [28], can be replicated and validated within an Iranian context. Specifically, we aimed to evaluate the face, content, construct validity, and reliability of the Persian-translated NLSE instrument. The initial phase involved forward and back translation following the World Health Organization (WHO) guidelines. Ultimately, the Persian version of the NLSE instrument was validated and prepared for standardization.

### Participants

The study population encompassed all first- to fourth-year nursing students. Adhering to psychometric standards [29], we established sample sizes of 210 participants for exploratory factor analysis (EFA; 10 per item) and 250 for confirmatory factor analysis (CFA), the latter incorporating a 10% attrition buffer. Through convenience sampling, we recruited 460 participants via direct in-class contact and faculty announcements, ensuring accessibility and willingness to participate. Following the exclusion of 10 cases with incomplete questionnaires from the CFA cohort, the final analytical sample comprised 450 participants (EFA: n = 210; CFA: n = 240). All participants met the inclusion criteria, being actively enrolled nursing students in years 1–4 who voluntarily provided written informed consent. While convenience sampling aligns with the practical demands of validation studies, its inherent risk of selection bias may constrain sample representativeness and generalizability to broader nursing student populations.

## Data collection

After obtaining the ethical code, the researchers introduced themselves to the nursing faculty. They explained the study design and objectives to the students, as well as how to complete the questionnaires. The samples were collected from September 10 to December 7, 2024.

The following tools were used to collect data

**Demographic questionnaire of research participants.** The demographic questionnaire for research participants includes several key variables: age, gender, marital status, residence status, educational level, academic semester, nursing-related work experience, interest in the field of study, and desire to immigrate.

**NLSE instrument.** Yiin et al. (2024) developed the Nursing Learning Self-Efficacy (NLSE) questionnaire [28], building on the Learning Science Self-Efficacy (SLSE) tool created by Lin and Tsai (2013). Lin and Tsai's multi-dimensional self-efficacy instrument effectively assesses students' self-efficacy in various learning domains. It expands traditional self-efficacy dimensions in science education—such as knowledge, higher-order thinking, practical skills, and everyday applications—by adding a dimension of "scientific communication [30]." With permission from Lin and Tsai, Yiin et al. adapted the SLSE tool to fit better the learning experiences and competencies essential for nursing education, resulting in the NLSE instrument. This tool evaluates nursing students' self-efficacy across similar dimensions, emphasizing the importance of communication in nursing. The questionnaire's dimensions and items were refined through discussions with a specialized committee of two education faculty members and two nursing specialists, leading to a reliable and valid instrument. Initially, the NLSE included 32 items across five dimensions: conceptual understanding, higher-order cognitive skills, practical tasks, everyday applications, and nursing communications. Following exploratory factor analysis, the final version of the NLSE retained 21 items distributed among the five original factors, each containing three to six items. The factor loadings demonstrated strong structural validity, and the overall reliability coefficient was 0.95, with individual factor reliability values ranging from 0.85 to 0.94 [28]. The dimensions of the questionnaire are as follows:

**Conceptual Understanding**: 3 items measuring confidence in grasping nursing definitions and theories.
**Higher-Order Cognitive Skills**: 5 items assessing confidence in problem-solving and critical thinking.
**Practical work**: 6 items evaluating competence in using nursing equipment.
**Everyday Applications**: 3 items assessing confidence in applying nursing concepts to real-life situations.

Each item was scored on a 5-point Likert scale (1 = strongly disagree to 5 = strongly agree). Subscale maxima were calculated as follows:

- Conceptual Understanding: 15

- Higher-Order Cognitive Skills: 25

- Practical Tasks: 30

- Everyday Applications: 15

- Nursing Communications: 20

The total score range was 21–105 (summing all 21 items). Higher scores reflect stronger self-efficacy among nursing students [28].

## Psychometric testing

**Translation Process.** In the initial phase of tool preparation, we adhered to the WHO tool translation guidelines [31]. Two bilingual individuals independently translated the English questionnaire into Persian, documenting all equivalents. After selecting the most appropriate terms, a collective agreement was reached on the Persian version. Next, two bilingual individuals, unaware of the original version, independently back-translated the Persian version into English. We then

compared the original and back-translated versions, leading to the final Persian translation. Finally, after reviewing and obtaining approval from the tool's designer, consensus was achieved on the final version of the questionnaire.

**Face validity.** Face validity was assessed using qualitative and quantitative methods [29]. Face validity was assessed using qualitative and quantitative methods [32], whereby participants rated each item's comprehensibility on a 5-point Likert scale ranging from 1 ("not at all understandable") to 5 ("completely understandable"). Impact scores were subsequently calculated for each item.

**Content validity.** Following the establishment of qualitative content validity and subsequent revisions of the items based on feedback from nursing faculty [33], a quantitative assessment of content validity was performed using two methodologies: the Content Validity Ratio (CVR) and the Content Validity Index (CVI). For the CVR evaluation, nursing faculty members rated each item on a three-point scale: "essential," "useful but not essential," and "not essential." To assess the CVI, each expert selected from the following options: (1) not relevant, (2) somewhat relevant, (3) acceptable relevance, and (4) fully relevant. The numerical values for the CVR were determined using the Lawshe table [34].

**Construct validity.** The sample size for the Exploratory Factor Analysis (EFA) was set at 210 participants, adhering to the standard guideline of having ten participants per item for the 21-item NLSE instrument [35]. Sampling adequacy was confirmed with a Kaiser-Meyer-Olkin (KMO) value of 0.89, significantly exceeding the acceptable threshold of 0.6 [36]. Additionally, Bartlett's Test of Sphericity was statistically significant ($\chi2 = 1854.37$, $p < 0.001$), indicating that the data were suitable for factor analysis. Principal Axis Factoring with Varimax rotation was employed, retaining factors with eigenvalues greater than 1. Items with factor loadings exceeding 0.4 were accepted [37]. In this study, communalities for all items were above 0.60, reflecting adequate variance explained by the factors.

In the next step, confirmatory factor analysis was employed to assess the degree of alignment between the factor structure of the Persian version of the multi-dimensional self-efficacy instrument for nursing students and its original version. For designing the structural model and evaluating the fit of this model, the Amos 24 software was utilized. Factor loading, which measures the strength of the relationship between the factor and the observable variable/item, was used to determine the validity of the measurement model. Factor loadings above 0.3 were considered favorable, while those above 0.6 were deemed favorable. The fit indices for this questionnaire model, based on confirmatory factor analysis, included seven indexes as follows: the chi-square to degrees of freedom ratio ($\chi^2/df$) ≤ 3, the root mean square error of approximation (RMSEA) ≤ 0.08, the comparative fit index (CFI) ≥ 0.90, the Tucker-Lewis index (TLI) ≥ 0.90, the incremental fit index (IFI) ≥ 0.90, the parsimonious normed fit index (PNFI) ≥ 0.50, and the parsimony comparative fit index (PCFI) ≥ 0.50 [38–41].

## Scale reliability

In the present study, we employed both internal consistency (Cronbach's alpha) [42] and stability (test-retest method) to assess the instrument's reliability. In this study, acceptable values for Cronbach's alpha and omega were considered to be 0.7 [43]. In addition, to evaluate the questionnaire's stability using the test-retest method, 30 nursing students completed the questionnaire at two different time points, two weeks apart.

## Convergent and discriminant validity

To evaluate the convergent validity of the NLSE instrument, we analyzed Composite Reliability (CR) and Average Variance Extracted (AVE). CR values were determined using the formula:

where λi represents the standardized factor loadings and var(∈i) denotes the error variances of the indicators [44]. All CR values exceeded the threshold of 0.7, confirming internal consistency [45]. The AVE, computed as:

$$CR = \frac{\sum (\lambda_i)^2}{\sum (\lambda_i)^2 + \sum \text{var}(\epsilon_i)}$$

$$AVE = \frac{\sum \lambda_i^2}{\sum \lambda_i^2 + \sum \text{var}(\epsilon_i)}$$

was above the recommended 0.5 for all constructs, indicating that the latent variables accounted for sufficient variance in their respective indicators [44]. Furthermore, CR values were greater than AVE values, satisfying the criterion for convergent validity [45].

For discriminant validity, we compared the Maximum Shared Squared Variance (MSV) with the AVE. MSV, derived from the squared inter-construct correlations, was confirmed to be lower than the AVE for each construct, ensuring that the constructs are empirically distinct [46]. This approach aligns with the Fornell-Larcker criterion, which posits that discriminant validity is established when the square root of the AVE for each construct exceeds its correlations with other constructs.

## Statistical analysis

The data analysis was conducted using SPSS v.24 for descriptive statistics and preliminary analysis, while AMOS v.24 was employed for confirmatory factor analysis (CFA) and structural equation modeling (SEM). Maximum likelihood estimation was used for parameter estimation, with a significance threshold set at $\alpha = 0.05$. For CFA/SEM analyses, we evaluated multiple parameters against established benchmarks: standardized factor loadings (>0.5 considered good, >0.7 excellent), modification indices (>20 suggested potential model improvements), and standardized residuals (absolute values <2.58 considered acceptable). Model fit was assessed using conventional thresholds: $\chi^2$/df ratio (<3 acceptable), RMSEA (<0.08 acceptable), CFI, IFI, and TLI (>0.90 acceptable), PNFI, and PCFI (>0.5).

Reliability analyses followed standard criteria: Cronbach's alpha (>0.7 acceptable, >0.8 good), composite reliability (>0.7), and average variance extracted (>0.5). For test-retest reliability, ICC values (>0.75 good, >0.90 excellent) were calculated with 95% confidence intervals.

## Ethical considerations

All methods complied with applicable guidelines, including the Declaration of Helsinki. The Ethics Committee of Ardebil University of Medical Sciences granted ethical approval for this study (Ethical code: IR.ARUMS.REC.1403.174). Informed written consent was secured from all participants before their involvement in the study.

## Results

### Characteristics of the participants

240 nursing students participated in this study, with a mean age of 22.43 years (SD = 2.25). The sample consisted of a slight majority of females (52.50%), and the largest proportion of participants were second-year students (36.70%), as shown in Table 1.

### Face validity

During the face validity phase, students provided insightful qualitative feedback, which was collected and recorded. Based on their suggestions, it was discovered that items 18 and 20 had conceptual similarities and overlaps. Additionally, the students recommended changing the phrase "create a solution" in item 7 to "provide a solution" and replacing the word "stages" in item 9 with a more suitable term. Subsequently, the research team collaborated to implement these suggested modifications. As a result, the wording of items 7 and 9 was revised based on the feedback received. To address the overlap between items 18 and 20, the conceptual distinction between these two items was clarified by consulting the original instrument designer. Following this consultation, the translation of item 18 was revised to highlight the phrase "the nursing knowledge I have personally acquired," distinguishing it from item 20. The quantitative analysis of the overall face validity for all items revealed impact scores between 1.90 and 4.09, all surpassing the acceptable threshold of 1.5.

**Table 1. Demographics of the participants (N = 240).**

| | | Frequency | Percentage (%) | Mean ± SD |
|---|---|---|---|---|
| Gender | Female | 126 | 52.50 | – |
| | Male | 114 | 47.50 | – |
| semester | First-year | 65 | 29.10 | – |
| | Second-year | 88 | 36.70 | – |
| | Third-year | 46 | 19.20 | – |
| | Fourth-year | 41 | 17.10 | – |
| Age | | | | 22.43 ± 2.25 |

Std: Standard Deviation.

## Content validity

The evaluation of content validity for the Persian version of the NLSE yielded the following results: The CVR index, assessed by a panel of 10 evaluators, set a threshold of 0.62 for each item, indicating suitability at this stage. In this study, CVR values ranged from 0.8 to 1, regarded as acceptable. The CVI also identified an acceptable score range between 80% and 100% (Table 2).

## Construct validity

The sampling adequacy for the analysis was confirmed by a Kaiser-Meyer-Olkin (KMO) value of 0.87, while Bartlett's test of sphericity was significant ($\chi^2 = 3439.98$, df = 210, p < 0.001), indicating suitability of the data for factor analysis. Employing Principal Axis Factoring with Varimax rotation and guided by the scree plot, five latent factors with eigenvalues exceeding one were identified. The eigenvalues for these factors were 5.16, 3.69, 3.31, 2.14, and 1.09, respectively. Together, these five factors explained 73.43% of the total variance observed in the scale items (Table 3).

In the next step, we also used Confirmatory Factor Analysis (CFA) to evaluate the construct validity of the NLSE instrument. Fig 1 shows the Persian version of the NLSE structure, consisting of five factors: (1) conceptual understanding (3 items), (2) higher-order cognitive skills (5 items), (3) practical work (6 items), (4) everyday applications (3 items), and (5) nursing communications (4 items). The CFA results indicated that all items had factor loadings above 0.3 (all p < 0.001), confirming that no items needed to be excluded (refer to Supporting file 3. for detailed factor loadings). Furthermore, the relationships between dimensions and items were statistically significant, with T-values greater than 1.96. The proposed model demonstrated good fit across various goodness-of-fit indices: $\chi^2$/df = 2.48, RMSEA = 0.07, TLI = 0.91, CFI = 0.93, IFI = 0.93, PNFI = 0.72, and PCFI = 0.75 (see Table 4).

Fig 1 depicts a CFA model consisting of five latent constructs, each connected to a specific set of observed indicators. Single-headed arrows represent the standardized factor loadings, demonstrating the relationships between latent variables and their corresponding indicators. Double-headed arrows indicate correlations among the latent constructs, and circles represent error terms for each observed variable.

## Convergent and discriminant validity

Convergent validity was confirmed through composite reliability (CR) values exceeding 0.7 and average variance extracted (AVE) surpassing 0.5 across all factors, satisfying established psychometric thresholds. Consistent with standard criteria, CR exceeded AVE for every construct. Discriminant validity was established as maximum shared variance (MSV) values remained below corresponding AVE estimates for all latent factors, confirming distinct dimensionality (Table 5).

**Table 2. Content validity of the Persian version of the NLSE instrument.**

| Item | CVR | CVI |
|------|-----|-----|
| 1 | 0.80 | 1 |
| 2 | 1 | 1 |
| 3 | 1 | 1 |
| 4 | 1 | 1 |
| 5 | 1 | 1 |
| 6 | 1 | 1 |
| 7 | 1 | 1 |
| 8 | 1 | 1 |
| 9 | 0.80 | 1 |
| 10 | 1 | 1 |
| 11 | 1 | 1 |
| 12 | 1 | 1 |
| 13 | 0.80 | 0.90 |
| 14 | 0.80 | 1 |
| 15 | 1 | 1 |
| 16 | 0.80 | 1 |
| 17 | 0.80 | 1 |
| 18 | 0.80 | 0.80 |
| 19 | 0.80 | 0.80 |
| 20 | 0.80 | 0.80 |
| 21 | 0.80 | 0.80 |
| Total | 0.90 | 0.95 |

**Table 3. Extracted Factors in EFA (n = 210).**

| Factors | Eigenvalue | Percentage of Explained Variance | Cumulative Percentage of Variance |
|---------|-----------|----------------------------------|-----------------------------------|
| Conceptual understanding | 5.41 | 25.77 | 25.77 |
| Higher-order cognitive skills | 3.57 | 17.01 | 42.79 |
| Practical work | 3.35 | 15.99 | 58.79 |
| Everyday application | 1.84 | 8.79 | 67.59 |
| Nursing communication | 1.51 | 7.19 | 74.78 |

## Reliability

The internal consistency and descriptive statistics for the 21-item NLSE instrument were excellent, with an overall mean score of 91.25 (SD = 10.63), reflecting strong participant performance. The instrument exhibited outstanding reliability, demonstrated by a Cronbach's Alpha of 0.89, Omega of 0.82, and a re-test with ICC coefficient of 0.89, which confirms its stability over time. Statistical significance was noted at $P < 0.01$*. The NLSE comprises five dimensions, each accompanied by corresponding Cronbach's Alpha and re-test coefficients, as detailed in the table. All values reported for these dimensions were deemed adequate and acceptable, further reinforcing the instrument's reliability, as illustrated in Table 6.

## Discussion

The psychometric evaluation of the Persian version of the NLSE scale demonstrates its suitability for assessing self-efficacy among nursing students in Iran. This study translated and culturally adapted the NLSE instrument into Persian

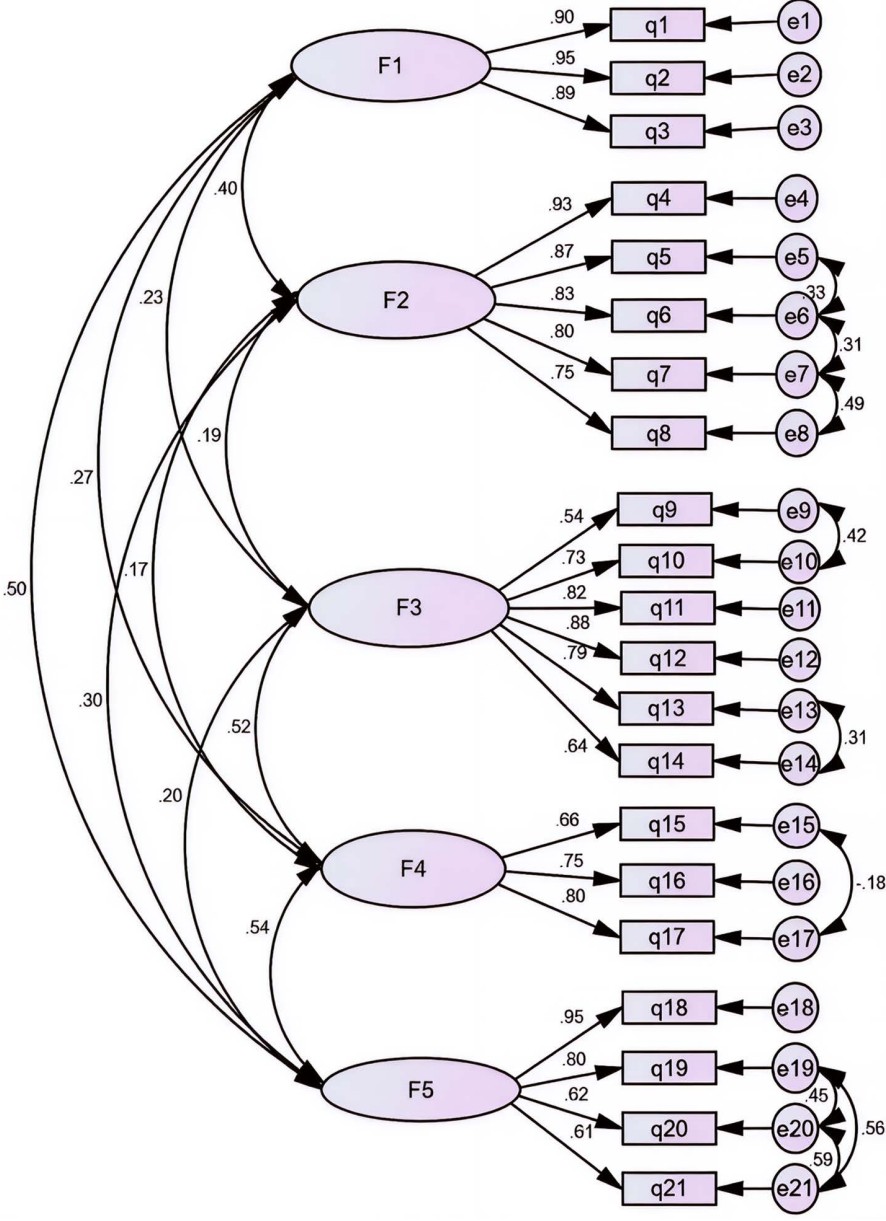

**Fig 1. The confirmatory factor analysis (CFA) model of the Persian version of the NLSE (n = 240).** F1: Conceptual understanding, F2: Higher-order cognitive skills, F3; Practical work, F4: Everyday application, F5: Nursing communication.

and rigorously evaluated its psychometric properties, aligning with best practices for instrument adaptation in populations with unique cultural, demographic, and linguistic characteristics [29,47].

The face validity assessment confirmed that the Persian NLSE items were clear, relevant, and culturally appropriate for Iranian nursing students. Qualitative feedback from students and quantitative impact scores, all surpassing the acceptable threshold [29], affirmed the instrument's face validity. In contrast to Yiin et al. [28], which relied on qualitative discussions with students and experts without reporting impact scores, the Persian adaptation employed a mixed-method approach, combining

**Table 4. Goodness-of-fit statistics for CFA models of the NLSE (*n* = 240).**

| Indices | Acceptable values | Fit index in the CFA model |
|---|---|---|
| χ²/df | ≤ 3 | 2.48 |
| RMSEA | ≤ 0.08 | 0.07 |
| CFI | ≥ 0.90 | 0.93 |
| TLI | ≥ 0.90 | 0.91 |
| IFI | ≥ 0.90 | 0.93 |
| PNFI | ≥ 0.50 | 0.72 |
| PCFI | ≥ 0.50 | 0.75 |

χ²/df: Chi-Square to Degrees of Freedom Ratio; RMSEA: Root Mean Square Error of Approximation;

CFI: Comparative Fit Index; TLI: Tucker-Lewis Index; IFI: Incremental Fit Index; PNFI: Parsimonious Normed Fit Index; Parsimony Comparative Fit Index.

**Table 5. Indices of the convergent and discriminant validity of the NLSE (*n* = 240).**

| Dimensions | CR | AVE | MSV |
|---|---|---|---|
| 1. Conceptual understanding | 0.93 | 0.83 | 0.25 |
| 2. Higher-order cognitive skills | 0.92 | 0.69 | 0.16 |
| 3. Practical work | 0.87 | 0.54 | 0.27 |
| 4. Everyday application | 0.93 | 0.54 | 0.29 |
| 5. Nursing communication | 0.84 | 0.57 | 0.29 |

CR: Composite Reliability; AVE: Average Variance Extracted; MSV: Maximum Shared Squared Variance.

qualitative insights with quantitative metrics. This dual approach provided a more robust evaluation of face validity, ensuring cultural relevance for the Iranian context. For instance, items were rephrased to align with Persian linguistic nuances, such as simplifying complex terms to match local educational terminology, enhancing comprehension among students.

Content validity was strongly supported by expert evaluations, with CVR and CVI scores exceeding recommended thresholds [29]. This indicates that the Persian NLSE effectively measures self-efficacy constructs relevant to nursing education in Iran. Yiin et al. [28] consulted four experts but did not report numerical CVR or CVI, limiting the transparency of their content validity assessment. By contrast, the current study's use of quantifiable measures, informed by a panel of ten Iranian nursing education experts, strengthens confidence in the instrument's content validity. Similar rigor is evident in the Persian adaptation of the Nursing Profession Self-Efficacy Scale (NPSES) [48], which also used expert panels to achieve high CVR and CVI, highlighting the importance of culturally informed expert judgment in Iranian psychometric studies.

Construct validity was thoroughly evaluated using a two-step factor analytic approach. EFA confirmed a five-factor structure—conceptual understanding, higher-order cognitive skills, practical work, everyday application, and nursing communication—accounting for a substantial portion of item variance. CFA further validated the model, with all factor loadings exceeding significance thresholds [29,49], indicating that each item meaningfully contributes to its corresponding construct. Fit indices from CFA demonstrated strong model fit, supporting the theoretical and statistical robustness of the Persian NLSE structure. This study incorporated CFA to confirm the measurement model, providing stronger evidence of construct validity in the Iranian context. Additionally, the Persian adaptation of the Self-Efficacy for Managing Chronic Disease (SES6G) scale [50] employed a similar EFA-CFA approach, confirming multidimensional structures relevant to Iranian populations, reinforcing the appropriateness of this methodology. Convergent and discriminant validity, assessed through composite reliability and mean-variance measures, exceeded recommended thresholds [49], aligning with Yiin et al.'s [28] findings in Taiwan and further affirming the Persian NLSE's ability to capture distinct nursing self-efficacy constructs.

**Table 6. Descriptive statistics of the 21-item NLSE (n = 240).**

| Dimensions | No. of item | Possible range | Total Mean ± SD | (α)Cronbach's Alpha >0.7 | Omega | ICC Coefficients* |
|---|---|---|---|---|---|---|
| Conceptual understanding | 3 | 1-6 | 13.14 ± 2.43 | 0.93 | 0.90 | 0.67 |
| Higher-order cognitive skills | 5 | 1-6 | 22.34 ± 3.58 | 0.93 | 0.88 | 0.63 |
| Practical work | 6 | 1-6 | 25.01 ± 4.62 | 0.88 | 0.88 | 0.76 |
| Everyday application | 3 | 1-6 | 12.73 ± 2.23 | 0.75 | 0.66 | 0.87 |
| Nursing communication | 4 | 1-6 | 18.01 ± 3.27 | 089 | 0.88 | 0.86 |
| Total (21-item NLSE) | 21 | 1-6 | 91.25 ± 10.63 | 0.89 | 0.84 | 0.89 |

Noticeable: $P < 0.01*$.

The Persian NLSE demonstrated high reliability, with strong internal consistency (Cronbach's Alpha and Omega) and test-retest stability, indicating consistent measurement of self-efficacy over time [49]. However, critical reflection reveals that certain subscales, such as nursing communication, exhibited slightly lower reliability (e.g., Cronbach's Alpha of 0.75 compared to 0.85–0.90 for other subscales). This may be attributed to cultural differences in communication behaviors, where Iranian nursing students may interpret communication-related items variably due to diverse clinical training experiences. For example, items addressing patient interaction may be less consistent in settings with limited patient contact early in training. Yiin et al. [28] reported comparable internal consistency but did not use the Omega index or discuss subscale-specific reliability, limiting insights into potential variability. The inclusion of Omega in this study, consistent with recommendations from recent psychometric literature [29], enhances the robustness of reliability evidence. Similar findings are noted in the Persian NPSES [48], where subscale reliability varied slightly due to contextual factors, suggesting that cultural nuances in nursing education may influence specific dimensions. The high test-retest reliability aligns with findings from the SES6G adaptation [49], underscoring the stability of self-efficacy instruments in Iranian settings.

The Persian NLSE's psychometric properties align with best practices observed in other Iranian psychometric studies, such as the NPSES [48] and SES6G [49], which emphasize rigorous translation, cultural adaptation, and comprehensive validity and reliability assessments. Unlike these studies, which focused on broader nursing or chronic disease contexts, the NLSE is uniquely tailored to nursing education, making it the first validated tool of its kind in Iran. Yiin et al.'s [28] original validation in Taiwan required minimal cultural adaptation due to linguistic homogeneity, whereas the Persian NLSE addressed significant linguistic and cultural differences, such as adapting items to reflect Iran's collectivist cultural norms and Islamic ethical frameworks in nursing practice. This adaptation enhances the instrument's applicability in Iran's unique socio-cultural context. The Persian NLSE fills a critical gap, as no comparable studies have adapted the NLSE for Iran. It can be distributed to nursing schools to assess students' self-efficacy, inform curriculum development, and identify students needing additional support. By addressing low self-efficacy through targeted interventions, the NLSE can enhance student engagement, clinical skills, and patient safety, contributing to a competent nursing workforce. Future studies should explore subscale reliability variations further, potentially refining communication-related items to improve consistency across diverse training settings.

## Limitations

One limitation of this study is its focus on a single nursing school in Iran, which restricts the generalizability of the findings to other student populations across various regions and cultural contexts. Additionally, since this instrument relies on self-reporting, its administration by instructors may introduce response bias among students. Another limitation is the relatively small sample size (n = 30) in the test-retest phase, which may limit the robustness of stability estimation. Including a larger number of participants in this phase could provide a deeper understanding of changes in students'

self-efficacy over time. Therefore, future research should assess this characteristic among students from diverse regions within the country. The instrument could also be adapted for use with students in other medical science disciplines beyond nursing.

## Conclusion

The psychometric evaluation of the 21-item Persian Nursing Learning Self-Efficacy (NLSE) instrument, encompassing five subscales—conceptual understanding, higher-order cognitive skills, practical work, everyday application, and nursing communication—confirms its robust validity and reliability for assessing self-efficacy among nursing students in Iran. Face and content validity, established through qualitative feedback and quantitative metrics, ensure the instrument's clarity and cultural relevance for the Iranian context. Construct validity, supported by exploratory and confirmatory factor analyses, verifies a clear five-factor structure, while high reliability, measured by internal consistency and test-retest stability, indicates consistent performance, though slight variability in the nursing communication subscale suggests potential cultural influences warranting further exploration. As the first culturally adapted tool of its kind in Iran, the Persian NLSE fills a critical gap in nursing education assessment. It enables educators to evaluate students' self-efficacy, enhance curriculum development, and identify those needing targeted support to improve clinical competence and patient safety. Future studies should apply the instrument across diverse nursing populations and settings, conduct longitudinal research to assess its predictive validity and responsiveness to educational interventions, and refine subscales to address cultural nuances, thereby enhancing its practical impact in nursing education..

## Supporting information

**S3 File. Standardized Factor Loadings.**
(DOCX)

## Acknowledgments

The authors extend their heartfelt thanks to all the students who participated in this study, the Student Research Committee of Ardabil University of Medical Sciences, and the Vice-Chancellor for Research at Ardabil University of Medical Sciences.

## Author contributions

**Conceptualization:** Elaheh Habibpour, Alireza Mirzaei, Mehraban Shahmari.

**Data curation:** Elaheh Habibpour, Reza Nemati-Vakilabad, Alireza Mirzaei, Nargess Ramazanzadeh, Sevda Gardashkhani, Mehraban Shahmari.

**Formal analysis:** Elaheh Habibpour, Reza Nemati-Vakilabad, Alireza Mirzaei, Mehraban Shahmari.

**Investigation:** Elaheh Habibpour.

**Methodology:** Elaheh Habibpour.

**Project administration:** Elaheh Habibpour.

**Resources:** Elaheh Habibpour.

**Supervision:** Mehraban Shahmari.

**Validation:** Elaheh Habibpour, Reza Nemati-Vakilabad, Alireza Mirzaei, Nargess Ramazanzadeh, Sevda Gardashkhani, Mehraban Shahmari.

**Visualization:** Elaheh Habibpour, Mehraban Shahmari.

**Writing – original draft:** Elaheh Habibpour, Reza Nemati-Vakilabad, Alireza Mirzaei, Nargess Ramazanzadeh, Sevda Gardashkhani, Mehraban Shahmari.

**Writing – review & editing:** Elaheh Habibpour, Reza Nemati-Vakilabad, Alireza Mirzaei, Nargess Ramazanzadeh, Sevda Gardashkhani, Mehraban Shahmari.

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
