## [Decision Letter · Decision Letter 0]

14 May 2025

Dear Dr. Shahmari,

Thank you for submitting your manuscript to PLOS ONE. After careful consideration, we feel that it has merit but does not fully meet PLOS ONE’s publication criteria as it currently stands. Therefore, we invite you to submit a revised version of the manuscript that addresses the points raised during the review process.

We look forward to receiving your revised manuscript.

Kind regards,

Mohammad Saadati

Academic Editor

PLOS ONE

Journal Requirements:

2. We have noted that the IRB document (IR.ARUMS.REC.1403.174) you have supplied does not match the approval number stated in your manuscript (IR.ARUMS.REC.1403.17). Please update the correct IRB number (IR.ARUMS.REC.1403.174) in the manuscript.

Reviewers' comments:

Reviewer's Responses to Questions

**Comments to the Author**

1. Is the manuscript technically sound, and do the data support the conclusions?

Reviewer #1: Yes

Reviewer #2: Yes

2. Has the statistical analysis been performed appropriately and rigorously?

Reviewer #1: Yes

Reviewer #2: Yes

3. Have the authors made all data underlying the findings in their manuscript fully available?

Reviewer #1: Yes

Reviewer #2: Yes

4. Is the manuscript presented in an intelligible fashion and written in standard English?

Reviewer #1: Yes

Reviewer #2: Yes

Reviewer #1: Dear editor

I am grateful the editor of the PLOS ONE journal for giving me the opportunity to review the article. I would also like to thank the research team for their valuable research. Below are some suggestions for improving the study process.

Introduction

1- Given that your research topic is in the field of self-efficacy, it is better to refer to Bandura's self-efficacy theory and write about the relationship of this theory to learning self-efficacy.

2- Have psychometric studies of the NILES instrument been conducted in other countries? Please list the psychometric results in other countries.

3- In Iran, a self-efficacy tool for clinical performance of nursing students has been developed and psychometrically tested. It is better to mention in the introduction the results of this study and how this tool differs from the tool you are studying. Did you only examine the students' theoretical learning self-efficacy in this study or was their clinical learning self-efficacy also examined?

4- It would be better to add this at the end of the introduction: "The psychometric properties of this tool, in addition to improving educational processes, can help predict and evaluate the success of students in completing coursework and academic progress."

5- At the end of the introduction, edit the sentence, instead of "... …multidimensional self-efficacy learning tool" write "…multidimensional learning self-efficacy tool..."

Methods

6- What was the type of study?

7- You also used exploratory factor analysis in this study. This validity is usually done after face and content validity. Please write the method in order and sequence.

8- What was the sample size for exploratory factor analysis and how did you determine it? What was the minimum acceptable factor loading for exploratory factor analysis? What indicators were used to conduct the analysis and what are their acceptable values? Reference should be given.

9- Please write the acceptable values of the fit indices used (CFI, RMSEA, PNFI, TLI, PCFI) in the CFA to confirm the model? References should be provided.

10-The acceptable Cronbach's alpha values for internal consistency should be written.

11- Have test-retest and omega been used for the reliability of the instrument, or has the ICC of the instrument also been examined? What are the acceptable values for the omega test?

Results

12- It would be better if the EFA results were also shown in the table, like the CFA results shown in Table 2, Because in this study you mentioned EFA in the method section but did not provide any data or information in the results section.

13- Why are there no omega reliability coefficient results in Table 4?

Discussion

14- In the discussion section, only one study (reference 17) has been used, which is the study of the original designer of the NILSE questionnaire. I think you should use similar studies in the field of instrument psychometrics and close to your study in the discussion section.

Reviewer #2: Dear Authors

Thank you for your valuable contribution. Your study provides important insights into the validation of a multidimensional instrument for evaluating nursing students’ self-efficacy in a Persian-speaking context. The methodology is generally sound, and the results are promising. However, several sections would benefit from greater clarity, conciseness, and refinement. Please consider the following comments for revision.

Title

Comment: The title is descriptive but slightly long. Consider simplifying it to improve readability (e.g., "Psychometric Evaluation of the Persian Nursing Learning Self-Efficacy Instrument").

Abstract

Comment 1: Consider adding a stronger concluding sentence to emphasize the implications or practical value of the instrument.

Comment 2: Avoid using abbreviations like PCFI or PNFI in the abstract without defining them.

Introduction

Comment 1: The introduction provides a comprehensive background, but the narrative could be more concise. Try to eliminate repetitive points about self-efficacy and academic performance.

Comment 2: Consider ending the introduction with a clearly marked paragraph stating the research aim(s) or hypotheses under a subheading such as “Objective”.

Methods

Study Design and Setting

Comment: The study is described as a methodological cross-sectional design, which is appropriate for psychometric evaluation. However, the absence of exploratory factor analysis (EFA) as a first step is notable. Even if CFA is justified due to prior structure (from Yiin et al., 2024), a rationale for not including EFA should be explicitly stated. This is especially important given the cultural-linguistic adaptation, which may impact factor structure.

Sampling and Participants

Comment 1: The authors report using a convenience sampling method, which is common in validation studies, but carries inherent selection bias. This limitation is acknowledged briefly in the discussion, but should also be addressed in the methods section with more detail about how it might affect generalizability.

Comment 2: The sampling size rationale (10:1 item-to-sample ratio) is standard and supported by citation. However, the final sample (n=240) exceeds the calculated requirement (n=210), which is positive. Still, there is no description of how the 250 students were approached or recruited, nor how non-responses or dropouts were handled.

Comment 3: Inclusion and exclusion criteria are mentioned, but they are vague. For example, the phrase "willing to participate" should be supported by more formal consent processes, and "incomplete questionnaire submissions" needs further clarification—were partial completions excluded entirely? How many?

Translation Process

Comment: The process followed WHO translation guidelines, which is appropriate and strengthens methodological rigor. The description is comprehensive and demonstrates cultural and linguistic sensitivity. It would be helpful to specify if any pilot testing or cognitive debriefing was conducted with the translated items before formal psychometric evaluation.

Validity Assessments

Face Validity:

Comment: The face validity assessment includes both qualitative (interviews) and quantitative (impact score) methods, which is excellent. However, the process for integrating feedback from students and the expert panel is not fully detailed. What kind of revisions were made?

Content Validity:

Comment: CVR and CVI assessments were well-executed, using Lawshe’s method and relevance rating scales. However, authors should report CVR/CVI values item-by-item in supplementary tables, not just as a range (e.g., “0.8 to 1”).

Construct Validity:

Comment: The use of CFA alone is limiting. As mentioned, EFA could have offered data-driven insights into whether the original structure is fully retained in the Persian context. CFA indices are well reported (χ²/df, RMSEA, CFI, etc.) and acceptable, but a visual representation of the factor loadings (e.g., a table or path diagram) would improve transparency.

Reliability Testing

Comment: The study used Cronbach’s Alpha, Omega coefficient, and test-retest (ICC), which provides a robust assessment of reliability. Yet, the sample size for test-retest (n=30) is relatively small and could affect stability estimation. Justify this choice or consider discussing it as a limitation.

Convergent and Discriminant Validity

Comment: Composite Reliability (CR), Average Variance Extracted (AVE), and Maximum Shared Squared Variance (MSV) are reported and interpreted correctly. It is a strength that these indices are used instead of relying solely on traditional correlation methods. However, formulas or calculation details for these measures could be briefly included or referenced to support replicability.

Statistical Analysis

Comment 1: The software used (SPSS and AMOS v.24) is standard. The description of statistical procedures is mostly accurate, but overly general. The structural equation modeling (SEM) is mentioned, yet no model parameters, modification indices, or residuals are discussed.

Comment 2: The study uses p < 0.05 as a threshold for significance, but does not report confidence intervals for ICC, factor loadings, or fit indices—these should be added for more rigorous interpretation.

Results

Comment 1: Some tables (e.g., descriptive stats) include excessive decimal places. Limit precision to two decimal places where appropriate.

Comment 2: Add a brief narrative description of Figure 1 to enhance reader interpretation of the CFA model.

Discussion

Comment 1: The discussion summarizes findings appropriately but could benefit from more critical reflection. For instance, why might certain subscales have lower reliability?

Comment 2: Compare your findings more directly with the original NLSE validation study by Yiin et al., particularly regarding cultural adaptation differences.

Comment 3: Consider restructuring the discussion to follow the order of your findings (face validity, content validity, construct validity, reliability) for better coherence.

Conclusion

Comment: The conclusion is aligned with the study’s findings. You may strengthen it by briefly suggesting practical applications (e.g., curriculum planning, student support services) and future research directions.

**Do you want your identity to be public for this peer review?** For information about this choice, including consent withdrawal, please see our Privacy Policy

Reviewer #1: No

Reviewer #2: No

---

## [Author Response · Author response to Decision Letter 1]

15 Jun 2025

Response to the Editor and Reviewers

We sincerely thank the Academic Editor and the reviewers for their constructive and insightful comments, which have been invaluable in improving the quality of our manuscript. Each point raised has been carefully considered and addressed in detail. All revisions are clearly marked using Track Changes within the manuscript. In summary, we have revised the manuscript in accordance with the suggestions and recommendations provided. We are grateful for the time and attention devoted to our work during this revision process.

Best regards,

Corresponding author

---

## [Decision Letter · Decision Letter 1]

17 Aug 2025

Psychometric Evaluation of the Persian Nursing Students' Learning Self-Efficacy Instrument

PONE-D-25-02738R1

Dear Dr. Shahmari,

We’re pleased to inform you that your manuscript has been judged scientifically suitable for publication and will be formally accepted for publication once it meets all outstanding technical requirements.

Kind regards,

Mohammad Saadati

Academic Editor

PLOS ONE

Reviewer #2: All comments have been addressed

2. Is the manuscript technically sound, and do the data support the conclusions?

Reviewer #2: Yes

3. Has the statistical analysis been performed appropriately and rigorously?

Reviewer #2: Yes

4. Have the authors made all data underlying the findings in their manuscript fully available?

Reviewer #2: Yes

5. Is the manuscript presented in an intelligible fashion and written in standard English?

Reviewer #2: Yes

Reviewer #2: Dear Authors,

Thank you very much for your thorough and thoughtful revisions. I appreciate the time and care you devoted to addressing each of the comments provided.

I have carefully reviewed your revised manuscript along with your point-by-point responses. I am pleased to confirm that you have adequately addressed all the concerns raised. The revised version shows significant improvement in clarity, structure, and methodological transparency.

Congratulations on your diligent work. I believe the manuscript is now suitable for publication in its current form.

**Do you want your identity to be public for this peer review?** For information about this choice, including consent withdrawal, please see our Privacy Policy

Reviewer #2: No

---

## [Editor Report · Acceptance letter]

PONE-D-25-02738R1

PLOS ONE

Dear Dr. Shahmari,

I'm pleased to inform you that your manuscript has been deemed suitable for publication in PLOS ONE. Congratulations! Your manuscript is now being handed over to our production team.

Kind regards,

on behalf of

Dr. Mohammad Saadati

Academic Editor

PLOS ONE